# Study on the Structural Characteristics of Bird Necks and Their Static Motion Features in the Sagittal Plane

Jiajia Wang [1,2], Wenfeng Jia [1], Fu Zhang [1,*], Xiqiang Ma [1], Zhaomei Qiu [1], Zhihui Qian [2,*], Luquan Ren [2], Zhijun Guo [1] and Yakun Zhang [1]

1  College of Agricultural Equipment Engineering, Henan University of Science and Technology, Luoyang 471000, China; jjw@haust.edu.cn (J.W.); 190319041068@stu.haust.edu.cn (W.J.); maxiqiang@haust.edu.cn (X.M.); qzm@haust.edu.cn (Z.Q.); gzhj1970@haust.edu.cn (Z.G.); zhangyk@haust.edu.cn (Y.Z.)
2  Key Laboratory of Bionic Engineering (Ministry of Education, China), Jilin University, Changchun 130022, China; lqren@jlu.edu.cn
*  Correspondence: zhangfu@haust.edu.cn (F.Z.); zhqian@jlu.edu.cn (Z.Q.)

**Abstract:** The necks of birds that possess complex structures, graceful curves, and flexible movements are perfect natural motion actuators. Studying their structural features, mechanic characteristics, and motion rules can provide valuable references for imitating such actuators and motion functions artificially. Previous studies have analyzed the influence of two-dimensional motion geometric features and anatomical structure of the neck on motion efficiency and motion stability. However, the mechanism of motion flexibility from the perspective of neck structure has not been investigated. This study investigates the general law of the relationship between the structural parameters and motion characteristics of birds' necks using tomography technology and 3D reconstruction technology. The results show that the structural characteristics of geese and ducks are similar, and there are significant differences in joint motion characteristics. Geese obtains complex neck postures through active intervertebral joints and highly flexible facet joints and possesses higher neck flexibility than ducks. This study provides a generic measuring method for obtaining birds' cervical spinal vertebral structural dimensional parameters and offers a new theoretical concept for bionic robotic structural design and manufacture.

**Keywords:** bird; neck; structure; motion range; sagittal plane

## 1. Introduction

Birds are one of the most dominant vertebrate groups in the earth's ecosystem. A wide variety of birds (about 10,000 species) are found in plains, plateaus, deserts, and rainforests. A few studies have shown their evolution pattern [1–3], structure, and development [4,5]. After billions of years of evolution, birds have obtained a long S-shaped neck with a high degree of flexibility and athletic ability. The role of birds' necks in the movement of the cervical spine remains unexplained. However, the neck has been relatively rarely studied compared to the limbs and skulls.

The cranial end of a bird's neck is composed of the atlas, which forms a joint together with the occipital condyle. The neck ends at the chest, and the vertebrae have true ribs connected to the sternum. Anatomically, the number of cervical vertebrae in birds ranges from 9 to 25, with numbers between 14 and 15 are the most common [6,7]. Functionally, the neck of birds is more flexible in dorsal and lateroflexion movements, and the saddle-shaped structure can prevent significant axial rotation of the atlas/axial caudal vertebrae. The geese possess 17 to 19 cervical vertebrae in the S-shaped neck [8], among which the 8th–11th cervical vertebrae are the longest. The 2nd–5th and the 15th–16th cervical vertebrae have the typical abdominal ridge, and the abdominal ridges of the 6th–14th cervical vertebrae have the vascular groove.

Several methods, including surface morphology measurement, X-ray imaging technology, and reverse engineering modeling methods, have been reported to study the structural characteristics of the bird cervical spine [9]. Terray used three-dimensional surface geometry morphometry to reveal a typical modular structure of cervical vertebra bones by examining 187 cervical vertebra bones from 16 species of birds [10]. Kambic quantified the lateral flexion and axial rotation of the wild turkey's neck using a biplanar X-ray test and subsequent processing. They found that many axial rotations can occur in the atlas and axis at the back of the joints, and maximum lateral buckling occurred in different joints at different back abdomen buckling angles. The axial rotation and lateral buckling are strongly coupled [11]. Krings' research shows that the upper and lower cervical vertebrae of the barn owl (a species of owl) are characterized by a wide central canal and a short articular process, while the middle is characterized by a narrow central canal and a large articular process [9]. Katzir studied the relationship between head stability, body mass, and leg length of four species of herons perched on vertical vibrating perches [12]. Van der Leeuw analyzed the characteristics and rules of feeding and drinking movement of domestic chickens and geese, and the study showed that the cervical spine movement of domestic chickens followed the geometric principle of maximizing angular efficiency [13]. Most of these studies focused on a typical function of the bird's neck, such as the large rotation range of the owls' head and neck, the high-speed pecking of woodpeckers, the stability in the roosting of herons, and the optimization of the feeding efficiency of domestic chickens. However, there is a lack of research on the structural characteristics of a goose's neck.

At present, avian cervical motion research subjects mainly include dorsoventral flexion, lateroflexion, and rotation. Most of them used the cadaver experiment method. Krings studied the anatomical basis of the excellent head rotation characteristics of barn owls [14]. X-ray fluorescence fluoroscopy was used to obtain the natural neck posture of living and dead owls when their heads were rotated, and CT scanning was used to obtain the shape of a single vertebral body. The research shows that the rotational motion can be described as a combination of the yawing axis and the rolling axis motion. Kambic studied the motion range of the three-dimensional cervical joint along the cranial–caudal axis of wild turkey carcasses. They summarized the motion range of birds' neck into three regions: the cranial joint mainly performed ventral flexion, with a high degree of axial rotation and lateroflexion; the caudal joint is mainly dorsiflexion with low axial rotation motion and high lateroflexion motion; the axial rotational range of motion (RoM) of the intermediate joint is variable and exhibits low lateroflexion [11]. In addition, to achieve a complex neck posture, the overlap of facet joints is reduced to the extent that the joints are almost separated during axial rotation, and the axial rotation and lateroflexion present a strong coupling relationship. Some studies also have directly shown that cervical morphology is a strong predictor of joint motion patterns and that musculoskeletal movement is a combination of multi-degree of freedom movements [15,16].

To sum up, previous studies have analyzed the influence of two-dimensional motion geometric features and anatomical structure of the neck on motion efficiency and motion stability. However, the mechanism of motion flexibility from the perspective of neck structure has not been investigated. This paper analyzes the birds' necks' structural characteristics, motion characteristics including ventroflexion/dorsiflexion, and how the birds' structural characteristics influence the motion characteristics based on the structural characteristics, dynamic analysis, and analysis of birds' characteristics neck.

## 2. Materials and Methods

### 2.1. Subjects and Experimental Setup

Five live geese (Lu'an, China) and ducks (Luoyang, China) were bought from a farmer's market in Luoyang City, Henan Province. The geese originate from Dabie Mountain, Lu'an City, Anhui Province. It is one of the best geese breeds in China after long-term artificial breeding and natural domestication [17]. We selected the geese and ducks as the experimental objects due to their wide distribution, easy access, and typical vertebra

morphology. Five moderately sized geese with body weights ranged from 4.40 kg to 6.00 kg, and five moderately sized ducks with body weights ranged from 2.25 kg to 3.10 kg were adopted for the experimental investigation. The CT images (Somatom Definition AS, Siemens, Munich, Germany) of cervical spinal vertebrae (C1-C17) at 0.6 mm intervals of five geese and five ducks in max ventroflexion and max dorsiflexion position were obtained under anesthesia at the Third Affiliated Hospital of Henan University of Science and Technology. The anesthetic used Zoletil 50 (0.1 mL/kg). During the experimental data collection, a goose and a duck at a time were scanned. The experiments involved in this study were approved by the First Affiliated Hospital of Henan University of Science and Technology (No.20200625). Once the experiment was completed, we immediately released them, the breeding and storage were not involved.

The Dicom format files of geese's and ducks' necks were generated from CT scan and then processed by the Mimics software (V17) for three-dimensional (3D) structural parameter measurements.

### 2.2. Structural Parameters Measurement

Commonly, the geese necks consist of 17 cervical vertebrae [8]. In this study, we only chose the C3-C15 segments for investigation. This is because C1 is immobile relative to the head and C16, C17 are immobile relative to the scapula [11], and the morphology and structure of atlas (C1) and axis (C2) are vastly different from other cervical vertebrae. Similarly, the characteristic structural parameters of the ducks' C4-C11 segments were measured.

In order to characterize the structure of the cervical vertebrae, six typical parameters were defined and measured schematically, as shown in Figure 1. These parameters include: the height (centrum height, CH) and width (centrum width, CW) of the articular facets of centra; the width of the zygapophyses measured from their most lateral points (zygapophyseal width, ZW); the angle between the articular surfaces of the zygapophyses (zygapophyseal angle, ZA); the length of the vertebral body (vertebral height, VH) and the height from the middle of the centrum to the most dorsal spine/surface (centrum length, CL). Parameter measurements were carried out for both the cranial and caudal ends of the vertebra.

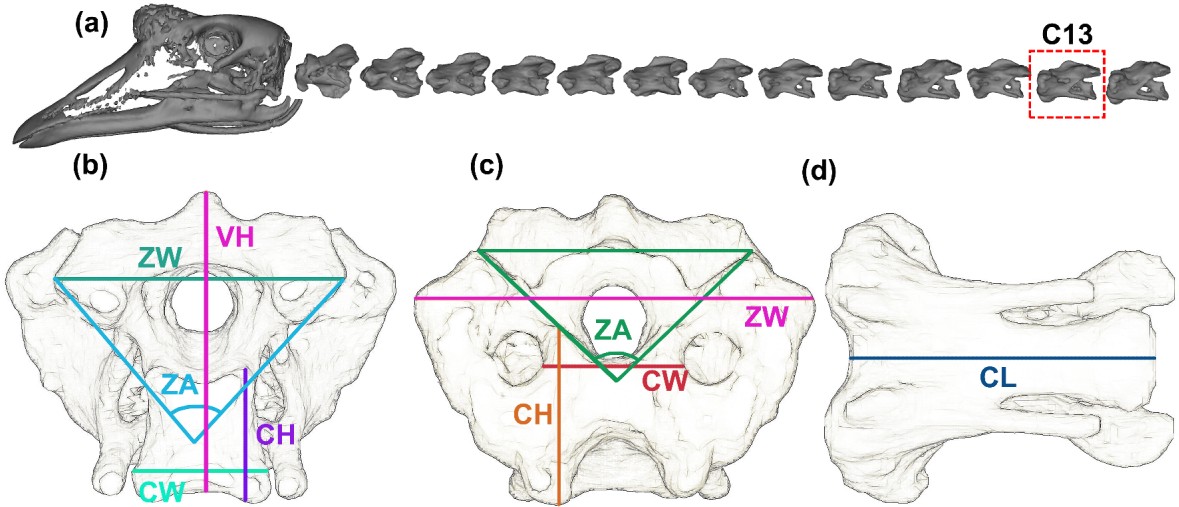

**Figure 1.** Tri-view measurement of the cervical spine. (**a**) C1-C14 cervical vertebrae. The front view (**b**), rear view (**c**) and side view (**d**) of C13. VH: vertebral height; ZW: zygapophyseal width; ZA: zygapophyseal angle; CH: centrum height; CW: centrum width; CL: centrum length.

We adopted the Mimics software to measure the above structural parameters. The measurements of each parameter were repeated three times. Then, the distance measurements are standardized by the cube root of individual body weight to account for the body size differenced [18]. The measurement error is about 1–2 mm. The error bars of these parame-

ters are obtained by calculating the mean value and the standard deviation of these data, and the variation rule of structural parameters is analyzed.

### 2.3. Motion Measurement of Articular Process Alignment

In order to obtain the motion characteristics of the adjacent facet of vertebrae for geese and ducks, we used Mimics software to measure the zygapophyseal overlap of the adjacent facets. The measurement error is about 1–2 mm. The measurement schematic diagram is shown in Figure 2, where Dz denotes the posterior articular process of the previous vertebra, and L is the anterior articular process of the latter vertebra. We normalized the measured data through formula (1). Thus, when the value is 100%, the facet is completely overlapping, and when the value is 50% or 150%, it is semi-overlapping.

$$Dz/L \times 100\% \tag{1}$$

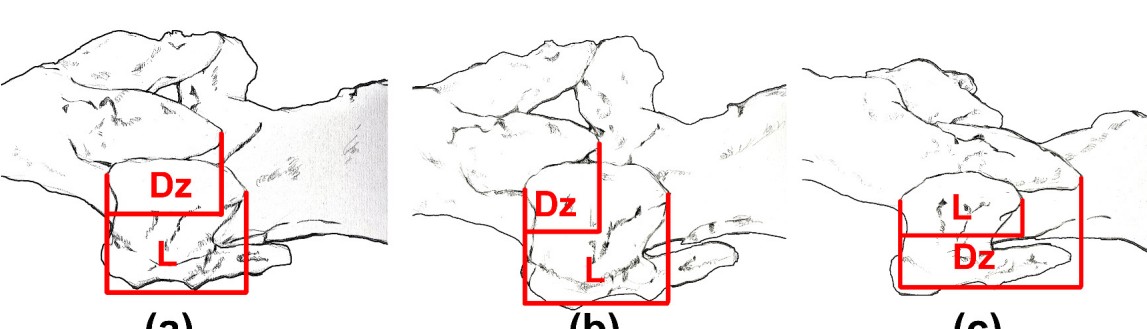

**(a)**   **(b)**   **(c)**

**Figure 2.** Schematic diagram showing the measurement of Zygapophyseal overlap for the adjacent facet. Dz: The amount of postzygapophyseal overlap with reference to the prezygapophysis. (L: The length of the prezygapophysis. (**a**) Two vertebrae in the normal state. (**b**) The state of the two vertebrae during ventroflexion. (**c**) The state of the two vertebrae during dorsiflexion.

### 2.4. Measurement of Ventroflexion/Dorsiflexion Motion

The 3D rendering model was obtained in Mimics. The sagittal plane was selected as the projection plane to obtain the position relationship of each cervical vertebra under the maximum ventroflexion/dorsiflexion movement, which was measured by AutoCAD 2014 (Autodesk, SanRafael, CA, USA).

The motion range of each joint in the ventroflexion and dorsiflexion postures was obtained, and the measurement schematic diagram is shown in Figure 3.

To measure the angle between adjacent cervical vertebrae, four points at the caudal end of the intervertebral foramen and four points at the cranial end of the intervertebral foramen are selected for each cervical vertebra (As shown in Figure 3a). The central blue point is calculated as the mean position of the four green points. The two central blue points at the caudal end and cranial end of the intervertebral foramen are connected to represent the transient position of each vertebra on the sagittal plane. Thus, the included angle between two adjacent vertebrae on the sagittal plane can be obtained, as shown in Figure 3b. The joint angle symbol is defined as "+" when the cranial vertebra is reversed to the ventral side with respect to the caudal vertebra, and "−" when the cranial vertebra is reversed to the dorsal side with respect to the caudal vertebra. Take the C7/C8 segments as an example, α represents the angle of C7 moves relative to C8, which is positive "+". On the other hand, take C12/C13 segments as an example, β represents the angle of C12 moves relative to C13, which is negative "−".

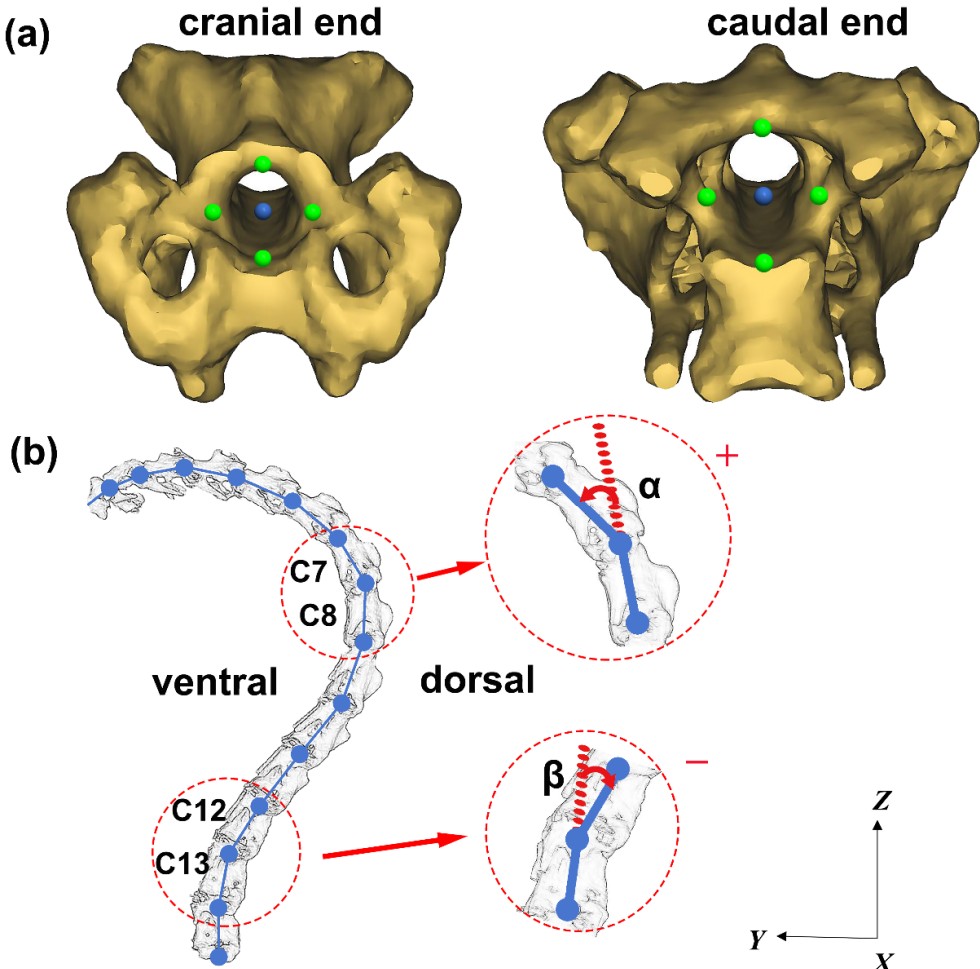

**Figure 3.** Schematic diagram for the ventroflexion/dorsiflexion motion measurement. (**a**) Representative positions of the selected four points (in green) and the calculated central point (in blue) at the caudal end and the cranial end of an intervertebral foramen. (**b**) Schematic diagram showing the included angle between two adjacent vertebrae.

## 3. Results

### 3.1. Structural Parameter Measurement

The cranial and caudal parameters, including ZW, CH, CW, VH, CL, and ZA of five geese and five ducks, were obtained. The measured parameters of the cervical spine of geese and ducks are shown in Figure 4. (Figure 4a,c) shows that the fluctuation of CH and CW of the cervical spine from the cranial end to the caudal end is not significant, while ZW shows a gradual upward trend from C7 to C13, then with a downward trend from C13 to C15. (Figure 4e) shows that CL gradually increases from C3 to C7 and then decreases from C10 to C14. However, the overall trend of VH is gradually decreasing. (Figure 4g) indicates that the cranial ZA shows a short decreasing trend from C3 to C6 and a gradually increasing trend after C10. The caudal ZA also shows a decreasing trend from C3 to C6, similar to the cranial ones, and then fluctuated gently.

As shown in (Figure 4b,d), the CH and CW of the cervical spine from the cranial end to the caudal end gradually increases from the middle part C7. In contrast, ZW shows a gradual upward trend from C6 to C11. (Figure 4f) indicates that CL gradually increases from C4 to C6 and then decreases from C6 to C11. VH barely changes from C4 to C7, and then decreases from C7 to C8, then increases weakly from C8 to C11.

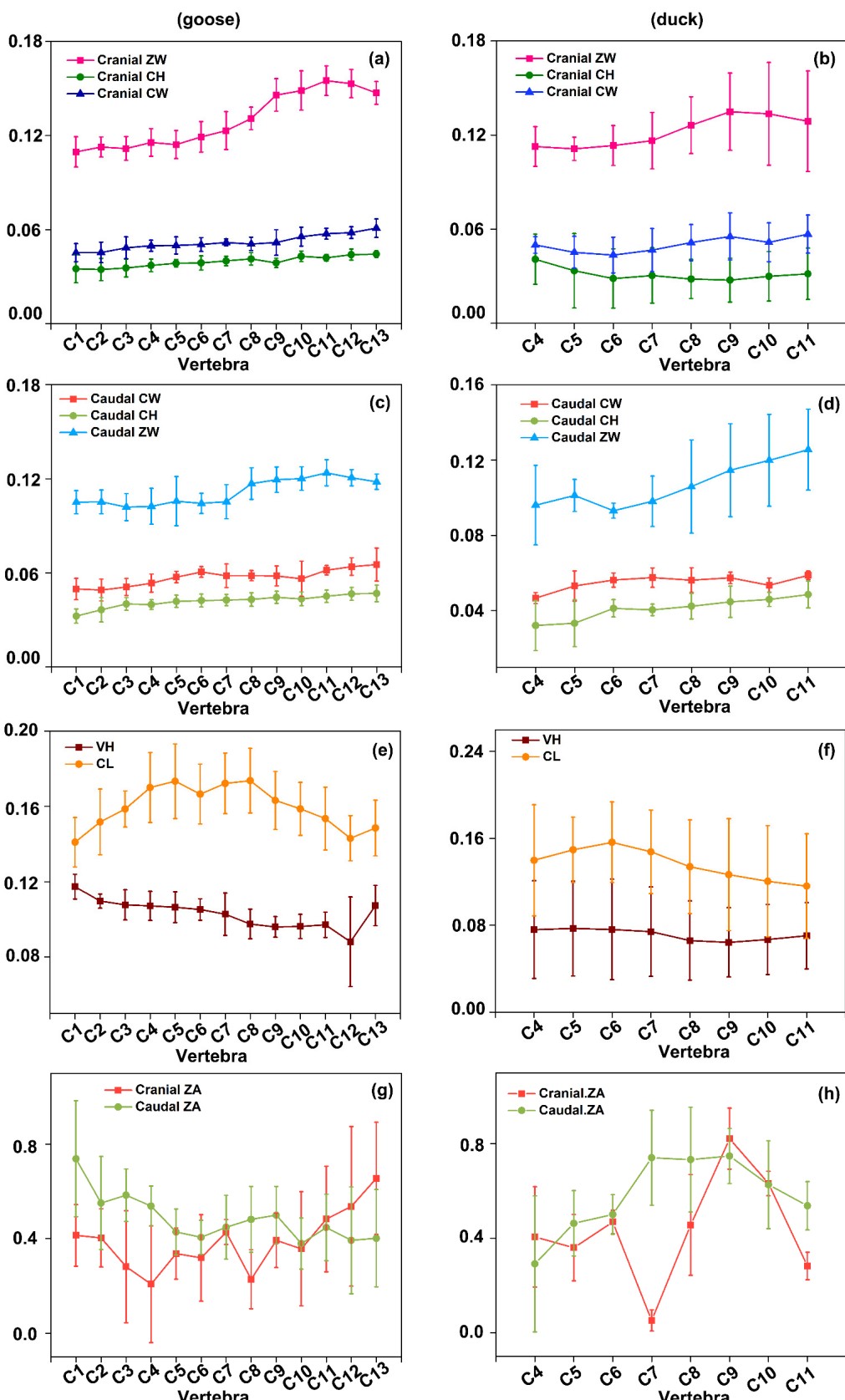

**Figure 4.** The linear and morphological angle measurements of goose and duck, Standardized processing of the original data, Mean and standard deviations are plotted. (**a**,**b**) ZW, CH, CW of Cranial of goose and duck. (**c**,**d**) ZW, CH, CW of Caudal of goose and duck. (**e**,**f**) VH and CL of goose and duck. (**g**,**h**) ZA of Cranial and Caudal of goose and duck.

### 3.2. Analysis of the Zygapophyseal Overlaps of Vertebrae Joints

The Zygapophyseal overlaps of several vertebral joints of five geese and ducks are calculated for maximum ventroflexion/dorsiflexion. The results are depicted in Figure 5, which shows that the overlap degree of geese increases from 50% (C3/C4) to 120% (C11/C12), then it decreases gently from C11/C12 to C15/C16. (Figure 5b) shows that the articular facet Zygapophyseal overlap increases from C5/C6 to C8/C9 in ducks. According to the contrast between geese and ducks, it is found that the geese have semi-overlap from C3/C4 to C7/C8, while the ducks only have semi-overlap at the C8/C9 joint. The total overlap occurs between C8/C9 and C15/C16 in geese and all nine joints in ducks.

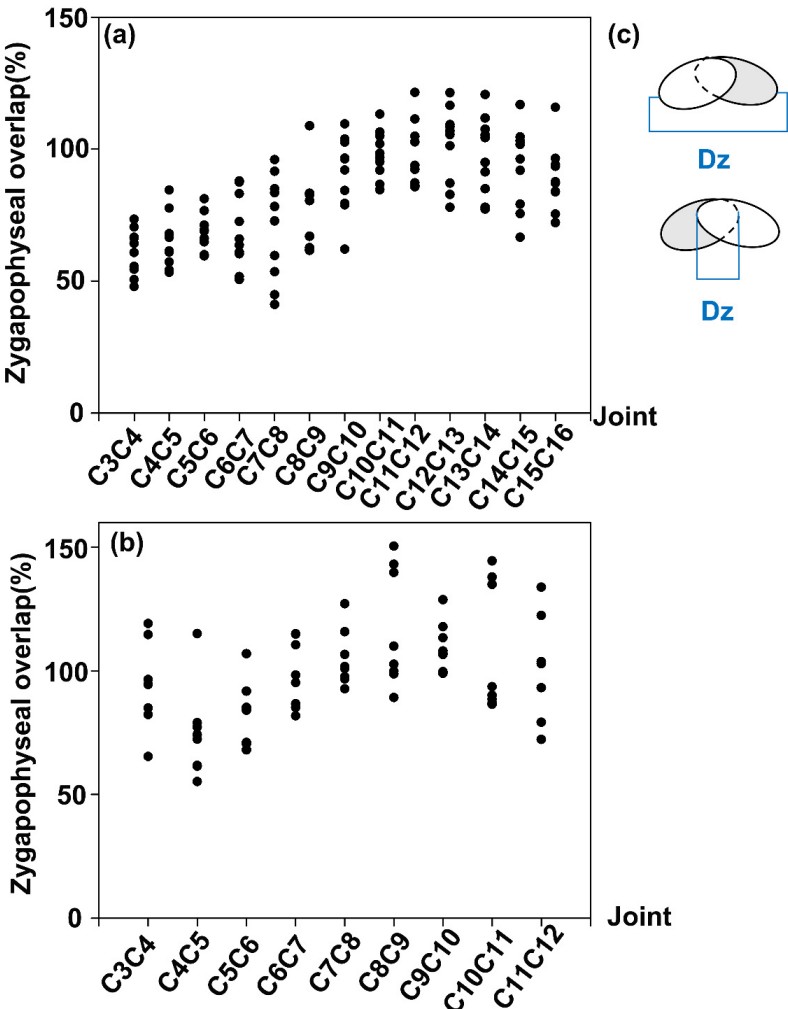

**Figure 5.** Schematic diagram of the vertebral Zygapophyseal overlaps in (**a**) geese and (**b**) ducks; (**c**) Zygapophyseal overlap diagram.

### 3.3. Ventroflexion/Dorsiflexion Movement Measurement

To study the range of motion (RoM) characteristics in ventroflexion/dorsiflexion movement for geese and ducks, we measured the maximum angle for every two adjacent vertebrae at the maximum ventroflexion posture and dorsiflexion posture, the overall RoM (defined as the difference between the angles at maximum dorsiflexion posture and ventrolflexion posture) in the sagittal plane, and the absolute mean RoM for each vertebral joint of five geese and ducks. The results are shown in Figure 6.

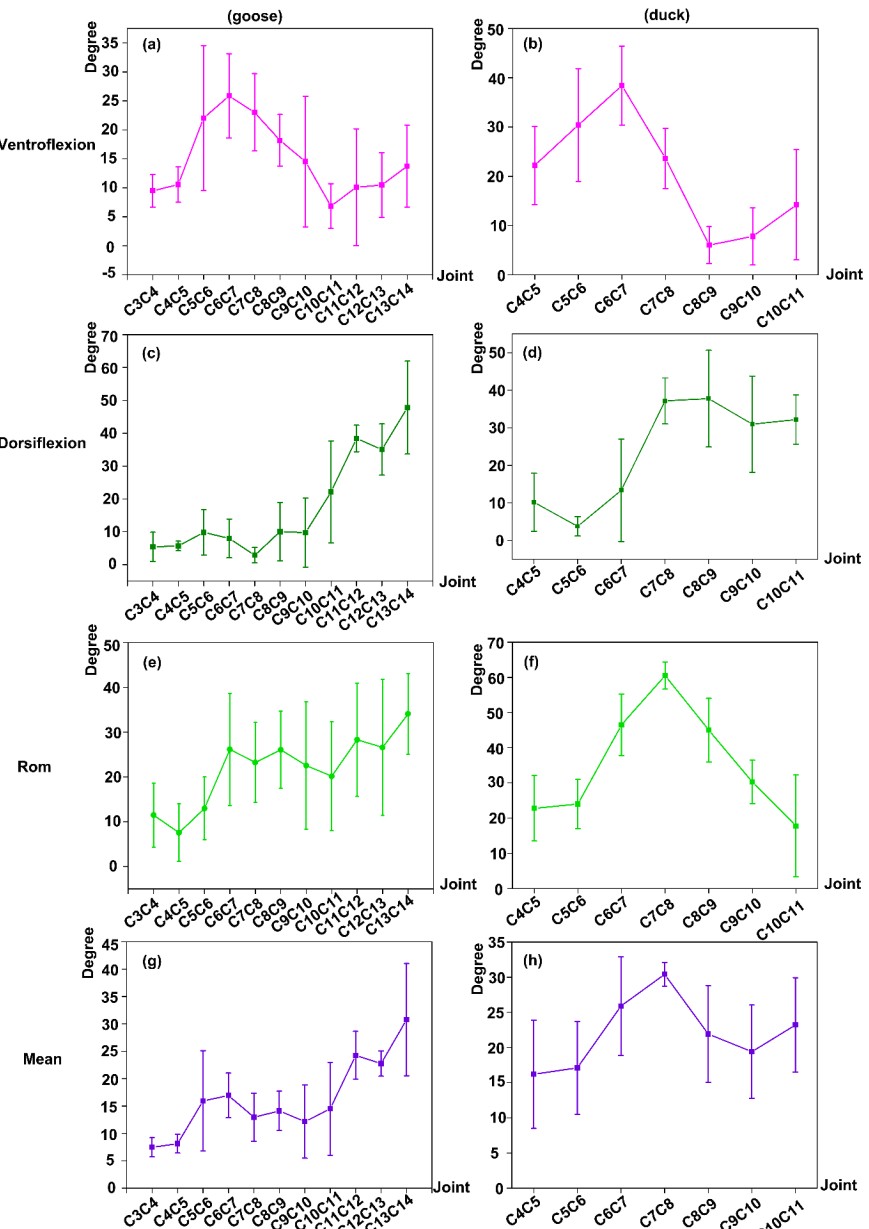

**Figure 6.** The RoM for goose and duck. (**a**,**b**) The measured maximum angle for adjacent vertebrae at the maximum ventroflexion posture. (**c**,**d**) The measured maximum angle for adjacent vertebrae at the maximum dorsiflexion posture. (**e**,**f**) The calculated RoM. (**g**,**h**) The mean-variance histogram of the absolute average value of ventroflexion and dorsiflexion movement. (Diagrams on the left are for geese, and diagrams on the right are for ducks).

### 3.3.1. Ventroflexion RoM

The maximum ventroflexion angle between adjacent vertebrae for geese and ducks was obtained from the calculation of each segment at the maximum ventroflexion posture. From the result in (Figure 6a,b), we can find that the overall vertebral motion trend of the geese and ducks is similar. The trends of both the two curves firstly increase and then decrease. Furthermore, the maximum motion angle occurs at C6/C7 joint for the geese and ducks. However, the minimum motion range occurs at C10/C11 for the geese and C8/C9 for the ducks. The results show that a large rotation occurs at C6/C7 joint for the geese and ducks when carrying out standard ventroflexion motion patterns, such as foraging and drinking activities.

### 3.3.2. Dorsiflexion Exercise

The maximum dorsiflexion angle between adjacent vertebrae for geese and ducks was obtained from the calculation of each segment at the maximum dorsiflexion posture. The results are shown in (Figure 6c,d), which clearly shows that that the joint angle curve of geese keeps unchanged from C3/C4 to C6/C7. After a slight decline at C7/C8, the curve continued to increase rapidly. For the ducks, the joint angle first decreases from C4/C5 to C5/C6, and increases rapidly from C5/C6 to C8/C9 rapidly, then decreases from C8/C9 to C10/C11. The results of dorsiflexion ROM of geese and ducks show that the maximum motion joint is in C13/C14, which is the most caudal joint for geese, and C8/C9 for the ducks. The results indicate that the geese's neck experienced a large rotation displacement from C11/C12 to C13/C14 joints during the dorsiflexion exercises, such as feather grooming. Similarly, there was a more significant angular displacement at joint C7/C8-C10/C11 for the ducks under the dorsiflexion exercise. Thus, the primary rotation segments are located at the caudal segments, either geese or ducks, when doing the dorsal stretch.

### 3.3.3. The Overall RoM in the Sagittal Plane

The overall RoM in the sagittal plane is equal to the absolute value of subtracting the ventroflexion RoM from the dorsiflexion RoM. The sagittal RoM mean-variance histogram of the geese and ducks is shown in (Figure 6e,f).

(Figure 6e) shows that the mean values of C6/C7-C8/C9 and C13/C14 joints of geese are more significant and have a more extensive range. C6/C7-C8/C9 belongs to the middle part of the geese's neck, which indicates that the high degree of motion in the middle part may be related to the ventroflexion movement of the geese when foraging and the dorsiflexion movement when grooming feathers.

The higher degree of motion around the C13/C14 joint may play a primary role in supporting the whole neck, which is needed for doing large ventroflexion, dorsiflexion, and other movements.

(Figure 6f) shows that the maximum motion RoM segment of ducks occurs at C7/C8, roughly consistent with geese. The results indicate that ducks perform larger vertebral flexibility in the middle of the neck, while geese perform larger flexibility in the caudal end.

### 3.3.4. The Absolute Mean Value of Ventroflexion and Dorsiflexion Exercise (Mean)

In order to further study the state of the vertebral structure of geese and ducks during ventroflexion/dorsiflexion exercise, the sum and average operation of ventroflexion data and dorsiflexion data of geese and ducks were carried out. (Figure 6g) shows that the geese reach a broad motion range at C5/C6 and C6/C7 joints. Then it gently declines and rapidly rises at C10/C11 joints. This indicates that the geese have more significant angular movement at several joints at the tail of the cervical spine during ventroflexion/dorsiflexion exercises. However, the ducks show more significant joint motion at the C7/C8 joint during the ventroflexion/dorsiflexion exercise, as shown in (Figure 6h).

It can be seen that there is a significant difference in the activity of cervical vertebrae between geese and ducks when they do ventroflexion/dorsiflexion exercise. In order to study the relationship between the cervical vertebral structure and neck movement of birds, we used ventroflexion, dorsiflexion, segmental RoM in the sagittal plane, and segmental mean motion of birds as the motion characteristic parameters.

We analyze the relationship between the structure and the movement of the experimental object based on the cranial and caudal vertebral structural characteristic parameters, such as cranial ZW(Cr.zw), Cranial CH(Cr.ch), Cranial CW(Cr.cw), Caudal ZW(Cau.zw), Caudal CH(Cau.ch), Caudal CW(Cau.cw), VH, CL, which was obtained in the experiments. The relationship between the cervical vertebral characteristic structural and motion parameters of geese and ducks is shown in Tables 1 and 2, respectively.

**Table 1.** The relationship between the cervical vertebral characteristic structural and motion parameters of geese.

| n | Direction of the Vertebrae | Cranial | | | | | | Caudal | | | | | | VH | | CL | |
|---|---|---|---|---|---|---|---|---|---|---|---|---|---|---|---|---|---|
| | Measuring Structure | CH | | CW | | ZW | | CH | | CW | | ZW | | | | | |
| | Vertebra | Cn | Cn+1 | Cn | Cn+1 | Cn | Cn+1 | Cn | Cn+1 | Cn | Cn+1 | Cn | Cn+1 | Cn | Cn+1 | Cn | Cn+1 |
| **3** | Ventroflexion | − | − | − | + | − | − | − | + | − | − | − | − | − | − | − | − |
| | Dorsiflexion | + | − | − | + | − | − | − | + | − | − | − | − | − | − | − | − |
| | Mean | + | − | − | − | − | − | − | − | − | − | − | − | − | − | − | − |
| | Rom | + | − | − | − | − | − | − | − | − | − | − | − | − | − | − | − |
| **4** | Ventroflexion | − | − | − | − | − | − | − | − | − | − | + | + | − | + | − | − |
| | Dorsiflexion | − | − | − | − | − | − | + | − | − | − | − | + | − | + | − | − |
| | Mean | − | − | − | − | − | − | − | − | − | − | − | + | − | + | − | − |
| | Rom | − | − | − | − | − | − | − | − | − | − | − | − | − | − | + | − |
| **6** | Ventroflexion | − | − | − | − | − | − | − | + | − | − | − | − | − | − | − | − |
| | Dorsiflexion | − | − | − | − | − | − | − | − | − | − | − | − | − | − | − | − |
| | Mean | − | − | − | − | − | − | − | + | − | − | − | − | − | − | − | − |
| | Rom | − | − | − | − | − | − | − | − | − | − | − | − | − | − | − | − |
| **7** | Ventroflexion | − | − | − | − | − | − | − | − | − | − | + | − | − | − | + | − |
| | Dorsiflexion | − | − | − | − | − | − | − | − | + | − | − | − | − | − | − | − |
| | Mean | − | − | − | − | − | − | − | − | − | − | + | − | − | − | + | − |
| | Rom | − | − | − | − | − | − | − | − | − | − | + | − | − | − | + | − |
| **8** | Ventroflexion | − | − | − | − | − | − | − | − | − | − | − | + | − | − | − | − |
| | Dorsiflexion | − | − | − | − | − | − | − | − | − | − | − | + | − | − | − | − |
| | Mean | − | − | − | − | − | − | − | − | − | − | − | + | − | − | − | − |
| | Rom | − | − | − | − | − | − | − | − | − | − | − | + | − | − | − | − |
| **9** | Ventroflexion | − | − | − | − | − | − | − | − | − | − | − | + | − | − | − | − |
| | Dorsiflexion | − | − | − | − | − | − | − | − | + | − | − | − | − | − | − | − |
| | Mean | − | − | − | − | − | − | − | − | − | − | − | + | − | − | − | − |
| | Rom | − | − | − | − | − | − | − | − | − | − | − | + | − | − | − | − |
| **10** | Ventroflexion | − | − | − | − | − | − | − | − | + | − | − | − | − | − | − | − |
| | Dorsiflexion | − | − | − | − | − | − | − | + | − | − | − | − | − | − | − | − |
| | Mean | − | − | − | − | − | − | − | + | − | − | − | − | − | − | − | − |
| | Rom | − | − | − | − | − | − | − | + | − | − | − | − | − | − | − | − |
| **11** | Ventroflexion | − | − | − | − | − | − | − | − | − | − | − | + | − | − | − | + |
| | Dorsiflexion | − | − | − | − | − | − | − | − | − | − | − | − | − | − | − | − |
| | Mean | − | − | − | − | − | − | − | − | − | − | − | + | − | − | − | + |
| | Rom | − | − | − | − | − | − | − | − | − | − | − | − | − | − | − | − |
| **12** | Ventroflexion | − | − | − | − | − | − | − | − | − | − | − | − | + | − | − | − |
| | Dorsiflexion | − | − | − | − | − | − | − | − | + | − | − | − | − | − | − | − |
| | Mean | − | − | − | − | + | − | − | + | − | − | + | + | − | − | + | − |
| | Rom | − | − | − | − | − | − | − | − | + | − | − | − | − | − | − | − |
| **13** | Ventroflexion | − | − | − | − | − | − | − | − | + | + | − | − | − | − | − | − |
| | Dorsiflexion | − | − | − | − | − | − | − | − | + | + | − | − | − | − | − | − |
| | Mean | − | − | − | − | − | − | − | − | + | + | − | − | − | − | − | − |
| | Rom | − | − | − | − | − | − | − | − | + | + | − | − | − | − | − | − |

**Table 2.** The relationship between the cervical vertebral characteristic structural and motion parameters of ducks.

| n | Direction of the Vertebrae | Cranial | | | | | | Caudal | | | | | | VH | | CL | |
|---|---|---|---|---|---|---|---|---|---|---|---|---|---|---|---|---|---|
| | Measuring structure | CH | | CW | | ZW | | CH | | CW | | ZW | | | | | |
| | Vertebra | Cn | Cn+1 | Cn | Cn+1 | Cn | Cn+1 | Cn | Cn+1 | Cn | Cn+1 | Cn | Cn+1 | Cn | Cn+1 | Cn | Cn+1 |
| **4** | Ventroflexion | − | + | − | − | − | − | − | − | − | − | + | − | − | − | − | − |
| | Dorsiflexion | − | + | − | − | − | − | − | − | − | − | + | − | − | − | − | − |
| | Mean | − | + | − | − | − | − | − | − | − | − | + | − | − | − | − | − |
| | Rom | − | + | − | − | − | − | − | − | − | − | + | − | − | − | − | − |
| **5** | Ventroflexion | + | − | − | + | − | − | − | − | − | − | − | − | − | − | + | + |
| | Dorsiflexion | + | − | − | + | − | − | − | − | − | − | − | − | − | − | + | + |
| | Mean | + | − | − | + | − | − | − | − | − | − | − | − | − | − | + | + |
| | Rom | + | − | − | + | − | − | − | − | − | − | − | − | − | − | + | + |
| **6** | Ventroflexion | − | − | − | − | − | − | − | − | − | − | − | − | − | − | − | − |
| | Dorsiflexion | − | − | − | + | − | − | − | − | − | − | − | + | + | − | − | − |
| | Mean | − | − | − | + | − | − | − | − | − | − | − | + | + | − | − | − |
| | Rom | − | − | − | − | − | − | − | − | − | − | − | − | − | − | − | − |
| **7** | Ventroflexion | − | − | − | − | − | − | − | − | − | − | − | − | − | − | − | − |
| | Dorsiflexion | − | − | − | − | − | − | − | − | − | − | − | − | − | − | − | − |
| | Mean | − | − | − | − | − | − | + | − | − | − | − | − | − | − | − | − |
| | Rom | − | − | − | − | − | − | − | − | − | − | − | − | − | − | − | − |
| **8** | Ventroflexion | − | − | − | − | − | − | − | − | − | − | − | − | − | − | − | − |
| | Dorsiflexion | − | − | − | − | − | − | − | − | − | − | − | − | − | − | − | − |
| | Mean | − | − | − | − | − | − | − | − | + | − | − | − | − | − | − | − |
| | Rom | − | − | − | − | − | − | − | − | + | − | − | − | − | − | − | − |
| **9** | Ventroflexion | − | − | − | − | − | − | − | − | − | − | − | − | − | − | − | − |
| | Dorsiflexion | + | + | − | − | + | − | − | − | − | − | − | − | + | + | + | − |
| | Mean | + | + | − | − | + | − | − | − | − | − | − | − | + | + | + | − |
| | Rom | − | − | − | − | − | − | − | − | − | − | − | − | − | − | − | − |

Table 1 shows that most segmental vertebral (C3/C4, C4/C5, C6/C7, C8/C9, C9/C10, C10/C11, C11/C12, C12/C13, and C13/C14) movements rise as the value of structural parameters increases. Moreover, the relationship between vertebral structural and motion characteristic parameters of vertebral segments C5/C6, C7/C8 is not apparent.

Among the joint segments whose structural parameter curves perform the same trend with the motion parameters, Cr.cw, Cau.ch, Cau.zw, Cau.cw, CL, and VH, perform the same trend with the motion parameters. In contrast, the rest of the structural parameters are less noticeable.

Table 2 shows the relationship between interarticular structure and movement of duck vertebrae. The results shows that the values of most joint movements and structures increases and decreases in the same way for ducks, such as C4/C5, C5/C6, C6/C7, C8/C9, and C9/C10. Some joint movements have no apparent relationship with structures, such as C7/C8, C10/C11.

Among the joint segments with the same trend of increase and decrease, the performance of structural parameters Cr.CH, Cr.CW, Cr.ZW, Cau.ZW, Cau.CW, VH, Cl is closer to the increase and decrease of motion trend, while other structural parameters are less pronounced.

## 4. Discussion

In this paper, firstly, the medical image acquisition method was used to obtain the maximum ventroflexion/dorsiflexion postures of geese and ducks. Then, through medical image processing and the three-dimensional (3D) and reverse modeling methods, the detailed anatomical structural characteristics of cervical vertebra segments and the RoM characteristics of geese and ducks were obtained. Furthermore, the relationship between structural characteristics and motion characteristics was discussed. The results show that the structural characteristics of geese and ducks are similar, and there are significant differences in joint motion characteristics.

First of all, the cervical spine structure of geese and ducks can be divided into three regions: the anterior region, the middle region, and the end region, which is similar to turkeys [19]. The vertebrae width measurement of goose and ducks included the cranial part and caudal part. The geese anterior cranial width (C3–C7 segments) is nearly the same, and the middle region (C8–C13 segments) gradually increases to the maximum, while the end region (C14–C15 segments) decreases to a certain extent. The cranial width of the cervical facet of ducks is similar to the geese, showing the characteristics of wide in the middle and narrow at both ends. That is, the width curve of the anterior region (C4–C6 segment) is relatively flat, increases at the middle region (C7–C9 segment), and decreases at the end region (C10–C11 segment). As for the caudal width, the geese anterior region (C3–C9 segment) is nearly equal, the width curve rises in the C9–C10 segment, and gently at the end region (C10–C15 segment). After a slight fluctuation in the anterior region (C4–C6 segment), the width of the caudal vertebrae of ducks increases linearly and reaches the maximum in the mid-caudal region (C7–C11). For the length of the center of the cervical spine, the anterior region of geese gradually increases, while the middle region remained unchanged, and the end region decreases. The change tendency of the cervical vertebra center length of the ducks is similar to geese, showing the characteristic of long in the middle and short at both ends. For the height of the vertebrae, the goose shows the characteristic of being high in the middle and low at both ends. Different from geese, there was a slight difference in the height of each vertebra of ducks.

As for the apparent contrast, the geese have 17–19 vertebral segments, and ducks have 14–16 vertebral segments. By comparing the average value of various cervical vertebrae's characteristic parameters in geese, we can conclude that: the longest vertebrae is the C7, while the shortest vertebrae is the C3; the highest vertebrae is C3, and the lowest vertebrae is C14; the cranial widest vertebrae is C13, and the narrowest vertebrae is C3, and the caudal widest vertebrae is C13, and the narrowest vertebrae is C5. For ducks, the longest vertebrae is C6, while the shortest vertebrae is C11; the highest vertebrae is C6, and of the lowest vertebrae is C8; the cranial widest vertebrae is C9, and the narrowest vertebrae is C4; the caudal widest vertebrae is C11, and the narrowest vertebrae is C6. To sum up, the comparison between geese and ducks shows that the central length, the height of the vertebra, the cranial and caudal zygapophysis width of the geese are more significant than that of the ducks.

Secondly, the zygapophyseal overlap of goose and ducks was measured to study the characteristics of joint motion of geese and ducks. The results show that the zygapophyseal overlap of the geese's joints ranging from 50% to 100%, which means that the joint of geese can realize the process from half overlap to complete overlap. The zygapophyseal overlaps of ducks' joints range from 50% to 150%, realizing the process from half overlap to complete overlap and then half overlap. The zygapophyseal overlap of the cervical spine in turkey ranges from 0 to 200% in the six joints [11]. The results show that compared with turkeys, the geese had higher zygapophyseal overlap during ventroflexion/dorsiflexion movement, which may be related to its comparatively smaller and complex joint motion.

Thirdly, during maximum ventroflexion movement, the maximum RoM measurements of geese and ducks indicate that the larger RoM of geese occurred in the anterior and middle region joints (C5/C6–C9/C10). Similarly, the larger maximum RoM occurs in the ducks' anterior and middle joints (C4/C5–C7/C8). During dorsiflexion move-

ment, the geese's larger maximum RoM occurs in the middle and partial end regions (C10/C11–C13/C14), and the ducks' larger RoM occurs in the middle region and partial tail region (C7/C8–C10/C11). In addition, previous studies have found that longer vertebrae allow more dorsiflexion [20–24].

The cervical motion RoM characteristics of geese and ducks in the sagittal plane were investigated. The results show that the cervical RoM of geese fluctuates wildly. The motion RoM gradually increases from C3/C4 to C6/C7 in the anterior region and reached the maximum. The RoM of the C8/C9–C10/C11 in the middle region is also more significant, indicating that the middle region's flexibility is greater than that of the anterior region. After a slight decrease in C11/C12 at the end region, the RoM of the C13/C14 joint increases slightly. Overall, the end region of the geese is more flexible than the middle region. In addition, the RoM change of duck is pronounced. The RoM of ducks increases rapidly from the C5/C6 segment in the anterior region to the C7/C8 segment in the middle region. Then the RoM decreases slowly, presenting a noticeable inverted U-shaped curve. The results show that the RoM of the C7/C8 segment reached the largest for ducks. To sum up, the most flexible segment locates in the middle region for the ducks, while the most flexible segment locates in the end region for the geese.

Furthermore, the relationship between cervical vertebra morphology and motion RoM during maximum ventroflexion and dorsiflexion for geese and ducks was analyzed based on the experimental results. The analysis results show that some cervical vertebra morphological parameters of geese, such as Cr.cw, Cau.ch, Cau.zw, Cr.cw, CL, and VH, obviously affect the joint motion parameters discussed in the study. Meanwhile, for ducks, the cervical morphological structural parameters, such as Cr.ch, Cr.cw, Cr.zw, Cau.zw, Cau.cw, CL, and VH, also significantly affect joint motion parameters. Therefore, we find that the structural parameters, such as Cr.cw, Cau.zw, Cau.cw, CL, and VH, appear in the two groups of different avian experiments. Similarly, the previous study on the turkey cervical vertebra also showed several vertebral characteristics, including vertebral length, vertebral width, vertebral height, zygapophyseal angle, and zygapophyseal width, influenced the joint motion. Therefore, we may conclude that the demonstrated above characteristic structural parameters that probably influence the joint motion characteristics of birds are universal.

In the study, the structural and motion characteristics were obtained for five geese and five ducks under anesthesia, and the relationship between structure and motion characteristic parameters was explored qualitatively. However, this study also has some limitations. Firstly, due to ethical restrictions, we investigated the structure and movement characteristics of birds' necks under living anesthesia instead of using a cadaver test. As a result, the CT imaging and modeling method was adopted to measure the dimensional morphological parameters. Deviations using CT imaging devices may occur during the data collection, and this type of error range was ±0.6 mm. Secondly, when conducting the static measurements of ventroflexion and dorsiflexion for geese and ducks, the coupled motion of both was not considered for simplicity. We will investigate the coupled motions in the follow-up research. Thirdly, the static pose of birds' maximum RoM was set instead of the continuous movement patterns because the birds' actual motion is consistent, so the two-plane X-ray test could be used in the later stage to ensure the continuous study of motion further.

## 5. Conclusions

This study aimed to investigate the cervical vertebral structural and joint RoM of five geese and five ducks under anesthesia. Meanwhile, the relationship between structural and motion parameters was analyzed qualitatively. In the study, the necks of geese and ducks showed a traditional three-region structure. Among the three regions, the vertebra in the middle region presented a larger three-dimensional size and larger motion RoM. Then, the structural parameters which made a significant influence on the motion parameters

were found. The influence mechanism is that the larger the size, the greater the RoM is. This study lays a theoretical foundation for future bionic robot design and manufacture.

**Author Contributions:** Conceptualization, J.W. and F.Z.; methodology, J.W.; software, X.M., Z.Q. (Zhaomei Qiu); validation, Z.Q. (Zhihui Qian); formal analysis, Z.G.; data curation, Y.Z.; writing— original draft preparation, W.J.; writing—review and editing, J.W.; supervision, L.R.; All authors have read and agreed to the published version of the manuscript."

**Funding:** This research was funded by the National Natural Science Foundation of China (No.51905155), the Henan Provincial Science and Technology Research Project (No. 212102110207).

**Institutional Review Board Statement:** Not applicable.

**Informed Consent Statement:** Not applicable.

**Data Availability Statement:** The dataset supporting the conclusions of this article is available in the figshare repository, https://dx.doi.org/10.6084/m9.figshare.15169623.

**Conflicts of Interest:** The authors declare no conflict of interest.

**Ethical Approval:** All experiments were conducted in compliance with national guidelines and in accordance with the Guide for the Care and Use of Laboratory Animals. The study was approved by the Third Affiliated Hospital of Henan University of Science and Technology.

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
