# Peer review of "Study on the Structural Characteristics of Bird Necks and Their Static Motion Features in the Sagittal Plane"

_coatings, doi:10.3390/coatings11101228_

Round 1
Reviewer 1 Report
The paper provides a generic measuring method for obtaining birds’ cervical spinal vertebral structural dimensional parameters, and offers a new theoretical concept for bionic robotic structural design and manufacture. The writing of this paper is excellent and I believe it is worth pulishing in your prestigious journal " coatings". Accordingly, I recommend an acceptance directly without any revision needed.
Author Response
Thanks for the reviewer’s comments.
Reviewer 2 Report
I`m sending my suggestions for Authors:
Material and methods:
The number of permisson of Ethics Committee should be add to this section.
Results section:
"3.1. Structural parameter measuremen" please change as: "3.1. Structural parameter measurement"
Figure 6 - please change word: "Rom" as: RoM". This abbreviation should be unified within the whole text.
Author Response
Dear reviewer:
Thank you for your review.
Material and methods:
The number of permisson of Ethics Committee should be add to this section.
Response:Thanks for the reviewer’s comments. The number of permission of Ethics Committee has been added to the Material and Methods section as follows,
“The experiments involved in this study has been appoved by the First Affiliated Hospital of Henan Univerisity of Science and Technology (No.20200625).”
Results section:
"3.1. Structural parameter measuremen" please change as: "3.1. Structural parameter measurement"
Response:Thanks for the reviewer’s comments. We have revised the manuscript according to the reviewer’s suggestion.
Figure 6 - please change word: "Rom" as: RoM". This abbreviation should be unified within the whole text.
Response:Thanks for the reviewer’s comments. We have revised the manuscript according to the reviewer’s suggestion.
Reviewer 3 Report
- Page 3, Lines 116-119: Provide relevant references.
- Page 3, Figure 1: In (a), label the highlighted box. In the figure caption explain the expansion of abbreviations of structural parameters as shown in (b),(c) and (d).
- How many repetitions were done for the different measurements? Provide the details in the relevant methods section.
Author Response
Dear reviewer:
Thank you for your review.
- Page 3, Lines 116-119: Provide relevant references.
Response:Thanks for the reviewer’s comments. We have provided relevant references according to the reviewer’s suggestion.
- Page 3, Figure 1: In (a), label the highlighted box. In the figure caption explain the expansion of abbreviations of structural parameters as shown in (b),(c) and (d).
Response:Thanks for the reviewer’s comments. We have revised the manuscript according to the reviewer’s suggestion.
- How many repetitions were done for the different measurements? Provide the details in the relevant methods section.
Response:Thanks for the reviewer’s comments. We repeted three times for each measurements. We have provided the details in the relevant methods section as follows,
“We adopted the Mimics software to measure the above structural parameters. The measurements of each parameter were repeated three times. Then, the distance measurements are standardized by the cube root of individual body weight to account for the body size differenced.”